# Features of the Structure of Layered Epoxy Composite Coatings Formed on a Metal-Ceramic-Coated Aluminum Base

**DOI:** 10.3390/ma18153620

**Published:** 2025-08-01

**Authors:** Volodymyr Korzhyk, Volodymyr Kopei, Petro Stukhliak, Olena Berdnikova, Olga Kushnarova, Oleg Kolisnichenko, Oleg Totosko, Danylo Stukhliak, Liubomyr Ropyak

**Affiliations:** 1Department of Electrothermal Processes of Material Processing, E.O. Paton Electric Welding Institute of the National Academy of Sciences of Ukraine, 11, Kazymyr Malevych Street, 03150 Kyiv, Ukraine; vn@paton.kiev.ua (V.K.); demianov@paton.kiev.ua (O.K.); 2Department of Computerized Mechanical Engineering, Ivano-Frankivsk National Technical University of Oil and Gas, 15 Karpatska Street, 76019 Ivano-Frankivsk, Ukraine; liubomyr.ropiak@nung.edu.ua; 3Department of Computer-Integrated Technologies, Ternopil Ivan Puluj National Technical University, 56 Ruska Street, 46001 Ternopil, Ukraine; totosko@tntu.edu.ua (O.T.); stukhlyak_d@tntu.edu.ua (D.S.); 4Department of Physical and Chemical Research of Materials, E.O. Paton Electric Welding Institute of the National Academy of Sciences of Ukraine, 11, Kazymyr Malevych Street, 03150 Kyiv, Ukraine; omberdnikova@paton.kiev.ua (O.B.); oskushnarova@paton.kiev.ua (O.K.)

**Keywords:** multilayer coatings, cumulative detonation spraying, oxide coating layer, epoxy composites, temperature, microstructure, subgrain structure, nanoparticles, dislocation density, hardening

## Abstract

Difficult, extreme operating conditions of parabolic antennas under precipitation and sub-zero temperatures require the creation of effective heating systems. The purpose of the research is to develop a multilayer coating containing two metal-ceramic layers, epoxy composite layers, carbon fabric, and an outer layer of basalt fabric, which allows for effective heating of the antenna, and to study the properties of this coating. The multilayer coating was formed on an aluminum base that was subjected to abrasive jet processing. The first and second metal-ceramic layers, Al_2_O_3_ + 5% Al, which were applied by high-speed multi-chamber cumulative detonation spraying (CDS), respectively, provide maximum adhesion strength to the aluminum base and high adhesion strength to the third layer of the epoxy composite containing Al_2_O_3_. On this not-yet-polymerized layer of epoxy composite containing Al_2_O_3_, a layer of carbon fabric (impregnated with epoxy resin) was formed, which serves as a resistive heating element. On top of this carbon fabric, a layer of epoxy composite containing Cr_2_O_3_ and SiO_2_ was applied. Next, basalt fabric was applied to this still-not-yet-polymerized layer. Then, the resulting layered coating was compacted and dried. To study this multilayer coating, X-ray analysis, light and raster scanning microscopy, and transmission electron microscopy were used. The thickness of the coating layers and microhardness were measured on transverse microsections. The adhesion strength of the metal-ceramic coating layers to the aluminum base was determined by both bending testing and peeling using the adhesive method. It was established that CDS provides the formation of metal-ceramic layers with a maximum fraction of lamellae and a microhardness of 7900–10,520 MPa. In these metal-ceramic layers, a dispersed subgrain structure, a uniform distribution of nanoparticles, and a gradient-free level of dislocation density are observed. Such a structure prevents the formation of local concentrators of internal stresses, thereby increasing the level of dispersion and substructural strengthening of the metal-ceramic layers’ material. The formation of materials with a nanostructure increases their strength and crack resistance. The effectiveness of using aluminum, chromium, and silicon oxides as nanofillers in epoxy composite layers was demonstrated. The presence of structures near the surface of these nanofillers, which differ from the properties of the epoxy matrix in the coating, was established. Such zones, specifically the outer surface layers (OSL), significantly affect the properties of the epoxy composite. The results of industrial tests showed the high performance of the multilayer coating during antenna heating.

## 1. Introduction

Current developments in the industry for the design of various mechanisms and machines pose the problem of increasing their reliability and durability under various operating conditions by using composite materials and coatings. It is promising to use metal-ceramic [1,2,3] and polymer composites [4,5,6], both in the form of coatings [7,8,9] and structural elements [10,11].

Improving performance is mostly achieved by changing structural parameters in composite [12,13,14] materials for various functional purposes [15,16,17]. Various technological methods are used to form layered coatings: combined, including photolithography, cryogenic dry etching, and thermal evaporation [18]; formation of epoxy composite material with Al_2_O_3_ microfiller [19]; and production of multilayer filtering porous permeable materials by dry radial isostatic pressing of powder together with pore-forming material [20]. Also, to solve this problem while simultaneously expanding the functional characteristics of various products that are operated, in most cases, in extreme conditions, multicomponent composite materials [21,22,23] and coatings based on them with a controlled structural organization [5,24,25] have proven themselves well.

Epoxy composite materials containing various additives are used to develop the design of components and parts of special-purpose equipment [26,27,28]. Composite materials based on epoxy matrices often replace steels and alloys because they have many advantages, including excellent corrosion resistance, higher endurance and elasticity, and lighter weight [29]. Much attention is paid to the performance characteristics of materials and coatings based on them, which have high corrosion resistance [30,31,32], physical and mechanical properties [33,34,35], wear resistance [36,37,38,39], and adhesive strength [26,40,41,42]. In particular, the contact interaction of fiberglass and steel shells with an epoxy adhesive layer was investigated by mathematical modeling in [43]. The peculiarities of fatigue failure of rods with a fiberglass shell and a carbon fiber core based on epoxy matrices were investigated in [44]. It was found that the uneven distribution of stresses in the cross-section of the rod and the inequality of the elastic moduli of the matrix and fibers lead to delamination of the shell and the appearance of longitudinal cracks.

In addition, during the development of equipment, attention is paid to its resistance to external factors: ultraviolet irradiation [45,46] and electromagnetic field [47]. To ensure the creation of high-performance composites, fillers obtained using new technologies are used: carbon nanotubes [48], multilayer structures of carbon hybrid polymer nanocomposites [49], graphene oxide modified with 4,4′-oxydianiline obtained by covalent grafting reaction [50], etc. In particular, good results were achieved when using nanocarbon tubes in composites [51,52,53].

Paper [54] analyzed competitive methods for improving the performance properties of functional layers of flat heating elements and proposed the use of ion-plasma (magnetron) spraying of a dielectric or resistive layer on stainless-steel substrates. However, this method of applying layers of flat heating elements is characterized by low performance.

In order to achieve the multipurpose effects of strengthening the surfaces of parts, improving their tribological properties, and protecting them from corrosion, layered [55] and functionally gradient coatings [56,57,58,59,60] are increasingly used. Multilayer coatings are also used to prevent icing [61,62] and to provide superhydrophobic properties to the surfaces of equipment elements in the aerospace industry [63,64].

Oxide coatings are formed by the combined electron-beam method on dielectric materials [65] and plasma electrolytic oxidation on both compact aluminum cast and deformed alloys [66,67,68] and sprayed aluminum coatings [69,70]. These coatings can be used both independently and as part of multilayer coatings. Plasma-electrolytic oxidation of aluminum and its alloys has been the subject of numerous studies. In particular, papers [71,72,73] describe the basic mechanisms of the oxidation process, and papers [71,72,74,75] focus on the properties of the resulting oxide coatings and their impact on the performance properties of materials.

A very effective means of renovation of damaged thin-walled structures is the use of thin coatings with good adhesive bonding to the substrate. In particular, works [76,77,78] consider methods for applying such coatings and evaluating their effectiveness, while references [79,80,81] analyze the strength and mechanisms of degradation of coatings during operation.

An example is the operation of parabolic antennas in various climatic zones, namely in marine climates, at moisture condensation temperatures, frost, various precipitation conditions, etc. When the product is in “sleep” mode and in the non-operational position, frost, ice, and accumulated snow may form on the surface of the antenna. These factors increase the time required to prepare the transition to the parabolic antenna operating mode. These problems require the creation of a heating system for the working surface of the product. The use of surface-distributed resistive electric heaters is effective. In the design of such products, it is promising to use multilayer coatings with different layer functions: electrical insulation, electric heating, thermal, and protective. It is most technologically feasible to place such coatings on the back of the antenna. For example, coating layers are formed by spraying aluminum oxide and then applying epoxy composite layers. It should be noted that each of the formed coating layers performs different functions: electrical insulation, conductive (heating layer), thermophysical, and protective. The reliability of the system is ensured by the adhesive–mechanical interaction of the coating layers in the composite material at the interface in the matrix–filler system. Additives made of dispersed fillers that are close to the nanoscale provide the creation of transition layers. These layers occur at the interface on the surface of the filler and differ in structure and properties from the epoxy binder. Such layers are referred to as outer surface layers (OSL). The thickness of the outer surface layer can reach 200-400 μm. Depending on the content of the filler, the volume of these layers significantly affects the properties of the composites as a whole [82,83,84]. By adjusting the OSL parameters, the conditions for targeted control of the properties of such layers at the interface are provided and, as a result, regulate both the physical and mechanical properties, as well as the thermal and electrophysical characteristics [85,86] of composites as a whole [22,24,87,88,89]. It should be emphasized that the material in the OSL has characteristics that differ from those of the epoxy binder base material in terms of the degree of crosslinking [90]. These properties depend mainly on the number of physical units per unit volume of the binder at temperatures below the glass transition temperature of the matrix.

Researchers Duriagina and Tepla [91] have established differences between the properties of micro- and nanoparticles of oxides and note that the small size of nanoparticles leads to the emergence of new, unique functional properties of materials.

The use of nanosized fillers will cause the transition of the epoxy binder material to the OSL state, improving the electrical characteristics of the composite. Nanofillers have a high specific surface area. Their modifying effect at low concentrations (up to 1%) will cause the binder to enter the OSL state. By controlling the concentration of the filler in the composite, its technical characteristics can be purposefully adjusted [85,86].

Based on the above review of the literature, it was found that research aimed at creating gradient coatings, consisting of layers of spraying aluminum oxide on a base and applying subsequent layers of epoxy composites, is promising in the development of an antenna heating system, and important from both scientific and technical points of view [92]. To solve this problem, it is necessary to conduct a number of studies on the effect of their structural parameters, including nanostructural formations, on the physical, mechanical, and electrical characteristics of layered coatings.

The aim of this paper is to investigate the structural parameters of materials of metal-ceramic layers of spraying aluminum oxides and aluminum and formed epoxy composites, carbon fabric, and basalt fabric in the form of a multilayer coating and their influence on the performance characteristics for the formation of a resistive flat electric heater for parabolic antennas of large diameters.

In order to achieve this goal, it is necessary to solve the following tasks:–To develop a technology for applying a multilayer coating by a combined method that includes spraying metal-ceramic layers Al_2_O_3_ + 5% Al onto an aluminum substrate, applying an epoxy composite layer, containing aluminum oxide, forming a working layer of a heating element from carbon fabric (impregnated with epoxy resin), applying a layer of an epoxy composite containing chromium oxide and silicon oxide, followed by the formation of an outer layer of basalt fabric;–To establish the structural features and study the physical, mechanical, and electrical properties of the developed multilayer coating.

## 2. Materials and Methods

### 2.1. Research Materials

For spraying the first two layers of combined coatings on an aluminum base, a mechanical mixture of the original Al_2_O_3_ powder of the Amperit^®^740.000 brand (99.5% Al_2_O_3_; 0.3% Na_2_O; 0.05% Fe_2_O_3_; 0.1% SiO_2_), (Höganäs AB, Höganäs, Sweden) and 5% Al (A995, Model CNPC-Al100), (CNPC Powder China Co., LTD, Shanghai, China) was used (Figure 1). The interest in metal-ceramic (Al_2_O_3_ + 5% Al) was caused by the fact that its use provides a combination of high hardness, characteristic of Al_2_O_3_ [93,94], with high plasticity and thermal conductivity, characteristic of Al, in the composite material.

Fractional composition of powders: Al_2_O_3_ for spraying the first coating layer was 22 μm, for the second—60–150 μm; pure Al—no more than 60 μm.

The following components were used to make the composite:–Epoxy-diane binder CYD-128, CAS No 25068-38-6 (Zhengzhou Meiya Chemical Products Co., LTD, Zhengzhou, China);–Polyethylene-polyamines (hardener epoxy-diane binder) CAS No 68131-73-7 (Shanghai Kean Technology Co., LTD., Minhang, Shanghai, China);–Aluminum oxide (Al_2_O_3_) with a dispersion of 10–20 μm; –Aerosil (SiO_2_) with a dispersion of 100 nm (CAS No 112945-52-5 (Ningbo Samreal Chemical Co., LTD, Ningbo, China);–Chromium oxide (Cr_2_O_3_) with a dispersion of 100 nm, CAS No 1308-38-9 (Ningbo Samreal Chemical Co., LTD, Ningbo, China);–Carbon fabric with a thickness of 200 μm brand GG-200 (ANGELONI, Group S.r.l., Quarto d’Altino, Italy); –Basalt fabric with a thickness of 200 μm brand BWP-200 (Changzhou Jlon Composite Co., LTD, Taihu Rd, Changzhou, Jiangsu, China).–Aluminum oxide (Al_2_O_3_) grade F16 with a grain size of 1.2 mm (Luoyang Weixiang Abrasives Co., Ltd, Henan, China).

The multilayer coating was applied to a sheet base made of technical aluminum AD0 (1011) standard ISO 209:2024 [95] (Shandong Xinghuasheng Steel Group Co., Ltd, Shandong, China).

Manufacturers of materials used in the research declare compliance with quality standards [96] and environmental management [97]. 

The total thickness of the layered aluminum-based coating, formed from sprayed layers of metal-ceramic (400 μm) and applied layers of polymer-oxide composites, inside which a carbon fabric is placed, and the outer layer is a basalt fabric, was 1600–1700 μm.

### 2.2. Technology of Layered Coating Formation

To apply multilayer coatings to an aluminum base (Figure 2), a technological process was developed, the scheme of which is shown in Figure 3. Experiments on spraying metal-ceramic layers of multilayer coatings were performed on equipment for high-speed multi-chamber cumulative detonation spraying (CDS) [98,99], which was developed in the framework of cooperation between the E.O. Paton Electric Welding Institute of the National Academy of Sciences of Ukraine (Kyiv, Ukraine) and PLAZER LLC (Kyiv, Ukraine) (Figure 4).

The technological process of forming multilayer coatings on an aluminum substrate included the following main operations: 1. Degreasing the surface of the aluminum base; 2. Jet-abrasive treatment of the surface of the aluminum base with F16 grade aluminum oxide 1.2 mm grain size; 3. Applying the first layer of metal-ceramic coating Al_2_O_3_ + 5% Al; 4. Application of the second layer of metal-ceramic coating Al_2_O_3_ + 5% Al; 5. Mechanical cleaning of the metal-ceramic coating surface; 6. Jet-abrasive treatment of the coating surface; 7. Applying a layer of epoxy composite based on CYD-128 with Al_2_O_3_ content; 8. Formation of a heating element based on carbon fabric impregnated with epoxy composite resin 9. Application of an epoxy composite containing Cr_2_O_3_ + SiO_2_, followed by the formation of an outer layer of basalt fabric applied to an unapproved Cr_2_O_3_ + SiO_2_ composite; 10. The compaction of the coating layers of materials containing epoxy composite was carried out on an installation equipped with an elastic diaphragm (chamber). 11. Quality control of multilayer coating.

The epoxy composites were vacuum dried in a DZF-6050 electric vacuum drying oven (Jincheng Industrial, Jintan, Changzhou, China) for 10–12 min at a vacuum of 133 Pa. Next, the appropriate coating layers were formed. After that, the layered coating was heat-treated at 120 °C in the same DZF-6050 electric vacuum drying oven for 2 h.

The advantage of the developed method of high-speed multi-chamber cumulative detonation spraying [100,101] compared to the known ones (gas flame, plasma coating, laser and hybrid technologies, etc.) [102,103] is to obtain a positive technical effect by achieving high speeds and pressures of the pulsed gas flow and high productivity of powder spraying on the substrate. This effect is achieved by accumulating energy in specially designed chambers of the device (Figure 4) by combining detonation combustion products in these chambers and creating a multi-frontal and quasi-continuous flow of them. CDS involves the high-frequency generation of pulsed jets of combustion products. Due to the accumulation of energy from the cylindrical and annular combustion chambers, two pressure maxima (3.5 MPa) occur. The two-chamber design of the device provides an increase in the velocity of the detonation combustion products, which allows for a higher velocity of the powder material. The velocity of the combustion products in the nozzle reaches 1800 m/s, which ensures effective acceleration of the powder material to speeds of 1200–1600 m/s with a pulse frequency of 30 Hz. The CDS method ensures the formation of dense nanostructured coatings with high adhesion, cohesion, and crack resistance [9,17].

Such a level of fracture toughness is commensurate with that of carbosteels [104] and coatings based on them [105], where high-manganese steel, capable of deformation strengthening through microtwinning [106], is used as the binder, and hard refractory carbide or boride phases [107] fulfill the role of the reinforcing phase.

Before the first layer of metal-ceramic coating was applied, the aluminum substrate was subjected to abrasive blasting (Figure 5).

For the deposition of the first two layers of metal-ceramic coatings (Al_2_O_3_ + 5% Al), the mode was used at a gun speed of 1500 mm/min and a spraying distance of 55 mm to the surface of the sample substrate. The first layer with a thickness of 150 μm was sprayed in three passes, the second layer with a thickness of 250 μm—in six passes. The porosity of the first layer of metal-ceramic coating was up to 1% (pore size 0.8–1 μm), in the second—20–25% (pore size 5–10 μm).

The next layer was formed from a composite: epoxy-diane binder (CYD-128) and aluminum oxide (Al_2_O_3_) with a dispersion of 10–20 μm.

In the next technological step, a 180–200 μm thick carbon fabric heating resistive element was formed on the previous layer. An epoxy composition was applied to the previous layer. Next, the carbon fabric was laid out. In this case, it was impregnated with an epoxy composition. Next, the composite was applied to the fabric, and an outer protective layer of basalt fabric was formed and sealed using an elastic diaphragm.

The high technical and economic performance of the layered coating was achieved due to the structural compatibility of the sprayed aluminum oxide with the metal surface of the antenna. These characteristics are due to the formation of nanostructural formations during the formation of the corresponding layers of the composition. The created gradient multilayer coating provides high adhesive and electrical breakdown strength, thermal conductivity, and crack resistance during thermal cycling in the conditions of operation of products (parabolic antennas) in the open air.

### 2.3. Methods for Researching the Structure and Properties of Coatings

An X-ray diffractometer DRON-UM1 (LNPO “Burevestnik”, Leningrad, USSR) was used for phase structural analysis. The studies were performed in monochromatic CuK_α_ radiation. Light and raster scanning microscopy were also used. For image processing of epoxy composites, computer programs in the MathCAD system were used. Light microscopy was performed in polarized light.

The dislocation structure was studied using a JEM-200CX electron microscope (JEOL Ltd., Tokyo, Japan) by transmission electron microscopy (TEM) at an accelerating voltage of 200 kV.

Such research provided information about the fine structure (size of subgranules, particles of hardening phases, distribution of dislocations, etc.) and will allow us to establish the relationship between the structure and the properties of the coating material. The results of these studies will make it possible to assess the influence of structural and phase components on the structural strengthening of the material. In addition, this approach will also make it possible to assess the level and sign of local stresses in the system (Al_2_O_3_ + 5% Al).

The porosity of the coatings was determined using software for the ZEISS Axio Imager M2m metallographic optical microscope (ZEISS Group, Jena, Germany) using the Pro Imaging software package that was supplied with the microscope. The microhardness of the coatings was measured on transverse microsections using a Leco M400 Hardness Tester (LECO Japan Corporation, Osaka, Japan) with a load of 1 N diamond Vickers pyramid.

Experimental research on the electrical breakdown of an epoxy composite was carried out on flat samples (100 × 100 mm, 0.1–0.2 mm thick) using a homemade device developed at Ternopil Ivan Puluj National Technical University. The electrical breakdown strength of epoxy composites was evaluated in kV per mm of thickness. The specimen was placed between two electrodes to which a voltage was applied to determine the breakdown value. The thickness was preliminarily measured at the electrode location. The number of tests of one epoxy composite was at least 10 times. If there was a deviation of more than 15% between the values of the breakdown electrical strength of the epoxy composite, the number of breakdown tests was increased so that the experimental error was reduced to 7–10%.

Fracture tests on the coating (Al_2_O_3_ + 5% Al) were performed on flat specimens (100 × 6 mm, 2.2–2.6 mm thick) during bending around a cylindrical rod using a NOVOTEST IZGIB SHG-2 device (NOVOTEST, Dnipro, Ukraine).

The adhesion strength of the metal-ceramic coating layers to the aluminum base was determined by the adhesive method. On the end surface of cylindrical samples with a diameter of 0.01 m, made of aluminum alloy, after abrasive blasting, a layer of metal-ceramic coating was formed by cumulative detonation spraying (CDS). Two such samples with metal-ceramic coatings were coaxially glued with VK-31 glue, which was polymerized at a temperature of 175 °C for 90 min under a load of 1 MPa. Peel tests were performed using self-centering grips on a Universal testing machine AGS-X, 10N–10kN (Shimadzu Corporation, Kyoto, Japan). The adhesion value of the coating was calculated by the well-known formula σ = 4*F*/(π*d*^2^). Here, *F* is the peel force, N; π—constant number (≅3.14); *d*—diameter of the cylindrical sample, 0.01 m. According to the results of five tests of samples, the arithmetic mean value of the adhesion strength of the coating to the base was determined, the value of which was approximately 80 MPa. In this case, destruction by glue occurred.

The thickness of the coating layers applied to the aluminum base was measured on transverse microsections of the multilayer coating by the microscopic (optical) method (ISO 1463:2021) [108], with an accuracy of 1 μm, using a measuring optical-digital instrumental microscope IMCL-100x50 (LLC VTP “AS-MA-PRYLAD”, Svetlovodsk, Ukraine). The total thickness of the multilayer coating was measured with an accuracy of 5 μm with a micrometer MK-25 with a division value of 0.01 mm and a measurement range of 0–25 mm (Mikrotech, Kharkiv, Ukraine). The thickness of the carbon and basalt fabric was also measured with this MK-25 micrometer.

## 3. Results

As a result of our research, it was found that the first and second layers of the obtained coatings (Al_2_O_3_ + 5% Al) were δ = 100–150 μm thick (layer 1, Figure 6a) and δ = 200–250 μm thick (layer 2). The porosity of the coatings was up to 1% (layer 1) and 20–25% (layer 2). The porosity of the first two metal-ceramic coating layers (Al_2_O_3_ + 5% Al) differs due to the need to ensure reliable adhesion of the multilayer coating to the aluminum base, the metal-ceramic layers to each other, and to the third layer of epoxy composite. The microhardness (HV) at a load of 1 N is HV = 8900–10,520 MPa (layer 1) and HV = 7900–10,250 MPa (layer 2). The phase composition of the first and second layers is similar: Al_2_O_3_—69%; Al_2_O_3_—15–15.2%; Al—15.8–16%. The ranges of microhardness values of the first and second metal-ceramic layers, Al_2_O_3_ + 5% Al differ. This is due to the fact that the composite materials of the specified metal-ceramic layers consist of a soft aluminum matrix Al and a hard filler—aluminum oxide Al_2_O_3_, which are characterized by different hardness values, respectively, and have different grain size and porosity. The accuracy of the hardness measurement was ≅5%, and the standard deviation of the measurement results was about 15–17%.

It has been proven that the concentration of selected oxides—Cr_2_O_3_ (Figure 7a,c,e), Al_2_O_3_ (Figure 7b,d,f) and SiO_2_ (Figure 7f,h) up to 1 wt% per 100 wt% of binder (hereinafter the concentration of dispersed additives (oxide powder) is given in wt% per 100 wt% of binder) increases the breakdown electrical strength (*E*) by 1.3–1.5 times.

The occurrence of OSL (Figure 7a,b) prevents the creation of conductive channels in the composite, and when they occur, their length increases significantly, i.e., *E*. Increasing the amount of filler in the epoxy composite, when the binder material goes into the OSL state, significantly increases the electrical strength of the materials as a whole. It should be noted that the matrix material goes into the OSL state partially or completely at different concentrations of filler in the material (Figure 7c–h). At the concentration of Al_2_O_3_ (10–20 wt%), SiO_2_ (1–3 wt%), and Cr_2_O_3_ (10–30 wt%), the electrical strength increases to *E* = 31–32 kV/mm, *E* = 57–62 kV/mm, and *E* = 45–53 kV/mm, respectively. When filled (1–2 wt%) with modified aerosil (n-aminopropyl aerosil), the electrical strength increases to *E* = 32–55 kV/mm.

It has been proven that modification of the filler surface improves both the physical and mechanical [109] and electrical characteristics of epoxy composites [24] due to better component combination. If this assumption is true, then treating the filler surface with an epoxy binder solvent, such as acetone, will improve *E*. Studies have shown that removing the solvent during heating and combining such a filler with a binder in a composition with subsequent material formation improves the electrical characteristics of the coating material to *E* = 115–120 kV/mm. The indicated maximum values of electrical strength were observed for the additive oxides Al_2_O_3_, Cr_2_O_3_, and SiO_2_ in the range of 25–120, 40–100, and 2–10 wt% per 100 wt% of binder.

The introduction of nanodispersed fillers Al_2_O_3_, as well as Cr_2_O_3_ and SiO_2,_ into the composition of the epoxy composite affects the formation of OSL due to the physical interaction between the solid surface of the filler and the binder. On the surface of these fillers, there are active centers (exchangeable electrons, OH groups, defects in the crystal structure), which interact with the macromolecules of the epoxy binder. In this case, the macromolecules are presented in the form of domains. Depending on the number of active centers, the layer located near the solid surface of the filler interacts due to the emergence of physical nodes. Such nodes are stable up to the glass transition temperature of the epoxy matrix. Processing of OSL images around dispersed particles in polarized light is characterized by a brightness gradient. Brightness determines the degree of crosslinking in OSL. The binder material in the OSL state has increased mechanical characteristics during cyclic loading. Migration of physical units during such loading will increase the resistance to the formation of micro- and subsequently macrocracks, thus increasing the crack resistance, long-term strength, and dielectric stability of the epoxy composite, and, accordingly, the service life of the multilayer coating. The outer surface layers (OSL) differ in mechanical and dielectric characteristics, and the parameters of such layers do not change during thermomechanical loading. Our previous studies [85] have established an increase in the mechanical strength of materials that include material in the OSL state. The electrical strength of such materials increases from 80 to 120 kV per millimeter of epoxy composite thickness.

At the next stage of the work, the features of the fine structure of the first coating layer (Al_2_O_3_ + 5% Al) were studied by TEM (Figure 8): the nature of the distribution and density of dislocations (ρ) in the structural components, the parameters of the substructure (*d*_s_) and phase separation particles (*d*_p_), etc. It is the characteristics of the first layer that affect the adhesive strength with the aluminum base material. Its properties determine the performance of the antenna heating element.

It was found that a substructure with a subgrain size *d*_s_ = 0.1–0.6 μm (Figure 8a,b) was formed in the coating material in the presence of Al_2_O_3_ nanoparticles with a size *d*_p_ = 0.02–0.12 μm (Figure 8c,d). Dislocation density in the coating: ρ = 2–3 × 10^9^ cm^–2^ (surface layer of the coating, Figure 8a–d) and ρ = 5–6 × 10^9^ cm^–2^ (fusion line (F/L) of the coating with the substrate, Figure 8e); in the substrate material: ρ = 4–5 × 10^10^ cm^–2^ (F/L, Figure 8e) and in the underlying aluminum substrate material ρ = 2–3 × 10^10^ cm^–2^ (Figure 8f).

Fractographic studies of the fracture surface of the coating (Al_2_O_3_ + 5% Al) revealed that the fracture character is predominantly ductile (Figure 9) with the presence of dispersed pits of 0.5–3 μm (Figure 9a–c) and elements of a quasi-brittle component (Figure 9d). The size of the quasi-brittle fracture facets is 3–5 μm (Figure 9d). The nature of this fracture indicates that the coating material (Al_2_O_3_ + 5% Al) is not prone to brittle fracture, which will ensure high crack resistance of such layered coatings in the operating conditions of antennas.

Subsequently, experimental and analytical assessments of the influence of the structure on the strength properties of the first layer (Al_2_O_3_ + 5% Al) of the coating were carried out. The analytical evaluation of the main components of the structural strengthening (dislocation—Δσ_D_, subgrain—Δσ_S_, dispersion—Δσ_D.H._) of the coating material was performed according to [9,13,88]. Structural parameters such as dislocation density (ρ), subgranular (*d*_s_) and dispersed phase particle sizes (*d*_p_), and particle spacing (λ_p_) were taken into account. It has been analytically established that high values of the coating hardening indexes are provided by nanoparticles of phases and substructure (Table 1).

The range of scattering values (Table 1) is large, as it is determined by the size of subgrains, the size of dispersed particles of phases, the distances between particles, and the density of dislocations. We determined these sizes using transmission electron microscopy. Then we determined the strengthening (Subgrain, Dispersion, Dislocation), using the obtained data.

The formation of an outer layer containing basalt fabric provides several functions: electrical insulation, protection against ultraviolet radiation, and increased structural strength of the multilayer heater coating (Figure 10).

The porosity of less than 1% of the first metal-ceramic layer (Al_2_O_3_ + 5% Al) was selected to ensure its reliable mechanical adhesion to the aluminum base, and the porosity of 20–25% of the second metal-ceramic layer (Al_2_O_3_ + 5% Al) was selected to ensure reliable mechanical adhesion to the first metal-ceramic layer and to the third layer of epoxy composite containing Al_2_O_3_. Heat is effectively removed from the multilayer coating into the body of the working part of the parabolic antenna, which is made of aluminum alloy with high thermal conductivity, and dissipates this heat through a mirror (reflective) surface into the environment. Also, part of the heat is removed through the epoxy composite layer containing Cr_2_O_3_ and SiO_2_, and the outer layer, which is made of basalt fabric, into the environment. This design of the multilayer coating ensures uniform and effective heat dissipation during heating of the parabolic antenna.

The developed technological process for forming a multilayer coating is suitable for practical application. To check the operability, the multilayer coating was formed using the developed technology (Figure 3) on the back side of a parabolic antenna made of aluminum alloy and used for its heating. The results of industrial tests of a parabolic antenna with the developed multilayer coating showed its high operability (for 9 years of operation). During a visual inspection of the multilayer coating, using a magnifier with a 7-fold magnification, no defects were found in this multilayer coating, so these tests are continuing.

## 4. Discussion

The results of our research are in good agreement with the results of the studies obtained during the study of the influence of structural organization on the properties of materials made of metal-ceramic coatings applied to an aluminum base [9,13,17]. The effectiveness of using epoxy composites to create flat resistive heating elements has been proven [4,7].

The design of the flat resistive heating element developed by Duriagina et al. [54] and the technology of its manufacture are effective. The implementation of this heating system includes the application of conductive resistive layers on a dielectric base by spraying. The disadvantage of this heating system [54] is the difficulty of controlling the thickness of the heating element and the technological parameters of layer formation using the ion-plasma spraying method.

The approach to the development of heating elements proposed by the authors of this paper does not have these disadvantages. The spraying of two metal-ceramic layers ensures the strength of the adhesive joints with the surface of the product. It should be noted that the formation of epoxy composites on a smooth aluminum base creates negative effects. The aluminum substrate is not an active surface in relation to the epoxy composite because a natural oxide film is always present on its surface, which is characterized by a small wetting angle. The layers of the composite coating interact with the surface of the oxide film through adhesive bonds. During thermal cycling in the course of antenna operation, the adhesive interaction decreases, which in turn reduces the efficiency of the flat heating element. To improve the efficiency of the heating system, two layers are formed on the surface of the product. The first layer of aluminum is applied by high-speed multi-chamber cumulative detonation spraying. This layer has high adhesive strength to the aluminum base. The porosity of this layer is less than 1%. A second layer is applied to the formed aluminum layer using the same method, which has a porosity of 20–25%. Next, an epoxy composite layer is applied to form a flat heating element. In this case, in addition to adhesive interaction, the epoxy composite penetrates the pores of the metal-ceramic layer. After the coating is formed, the adhesive interaction increases due to the mechanical component. The magnitude of this interaction is determined by the strength characteristics of the epoxy composite. In this case, the adhesive strength is close to the cohesive strength of the epoxy material. The specified heating system, formed in the form of a multilayer coating, ensures stable operation of products during repeated thermal cycling and other external factors. It should be noted that the developed multilayer coating did not exhibit the phenomena described in [110,111] during its operation.

The design strategy of the multilayer coating consisted of a rational choice, a combination of physical and mechanical properties of functional layers, and their strong connection with each other. Since the research is conceptual and pioneering, and the optimization of geometric and physical parameters of a multilayer coating requires significant efforts and a lot of time, in this study, complex optimization was not carried out, due to the urgent need to solve the technical problem—the creation of a parabolic antenna heating system in a short period of time.

In the future, it is planned to research the processes and topological parameters during the crosslinking of epoxy composites and the effect of different methods of oxidation of the aluminum base on the operational properties of flat heating systems for thermal control of equipment elements, as well as conduct multi-criteria optimization of the thickness of the layers and their porosity.

## 5. Conclusions

As a result of the research, it was found that during the multi-chamber detonation spraying of the first and second layers (Al_2_O_3_ + 5% Al), layers with a thickness of δ = 100–150 μm and δ = 200–250 μm were formed with their average microhardness of 9710 MPa and 9075 MPa, respectively.

It is proven that the method of detonation high-speed spraying provides the formation of a subgrain structure with a size of 100–300 nm and phase particles with a size of 10–100 nm and their uniform distribution in the coating material. This helps to improve the strength of the coating material due to substructural and dispersion strengthening. The formation of nanocluster-sized particles in the metal-ceramic layers of a multilayer coating occurs due to the action of high energies during cumulative detonation sputtering on Al_2_O_3_ + 5% Al powder particles, which hit a solid, stationary aluminum base at high speed, contributing to the nucleation and growth of these phase particles.

The formation of a nanostructural state in the coating material (Al_2_O_3_ + 5% Al) with a uniform distribution of dislocation density prevents the formation of residual stress concentrators, which ensures high crack resistance of such coatings in the operating conditions of antennas.

The effectiveness of using nanofillers of aluminum, chromium, and silicon oxides in epoxy composites has been proven. The appearance of structures near the surface of these additives that differ from the properties of the coating matrix was revealed. Such zones (outer surface layers) significantly affect the properties of the epoxy composite. By scientifically predictable adjustment of the layer thicknesses, the required performance properties of a multilayer composite flat antenna heater were achieved.

## Figures and Tables

**Figure 1 materials-18-03620-f001:**
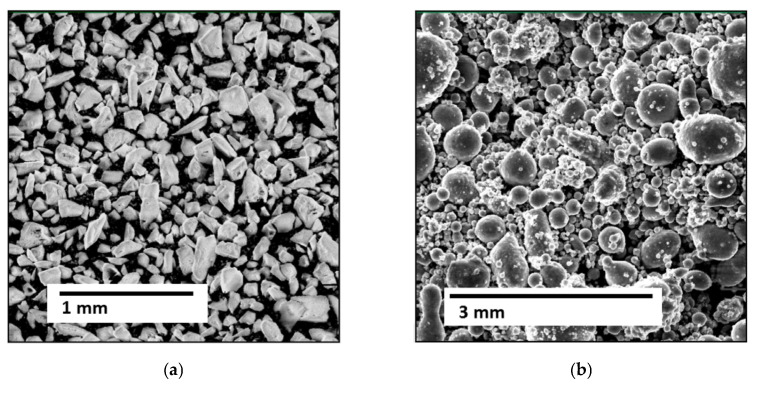
Photos of powders for spraying a metal-ceramic coating layer: (**a**) Al_2_O_3_ (×600); (**b**) Al (×300).

**Figure 2 materials-18-03620-f002:**
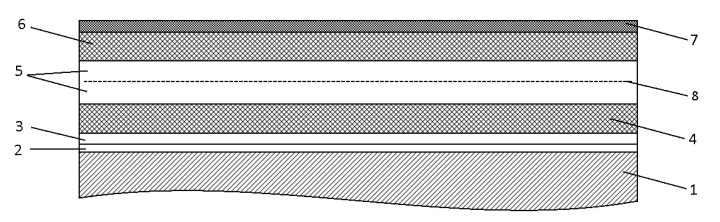
Schematic representation of a multilayer coating: 1—product base–substructure (aluminum sheet); 2—metal-ceramic layer Al_2_O_3_ + 5% Al (applied by high-speed multi-chamber cumulative detonation spraying, porosity less than 1%); 3—metal-ceramic layer Al_2_O_3_ + 5% Al (applied by high-speed multi-chamber cumulative detonation spraying, porosity 20–25%); 4—epoxy composite with Al_2_O_3_ filler; 5—heating layer (carbon fabric impregnated with epoxy resin); 6—epoxy composite with Cr_2_O_3_ + SiO_2_ filler; 7—basalt fabric; 8—carbon fabric.

**Figure 3 materials-18-03620-f003:**
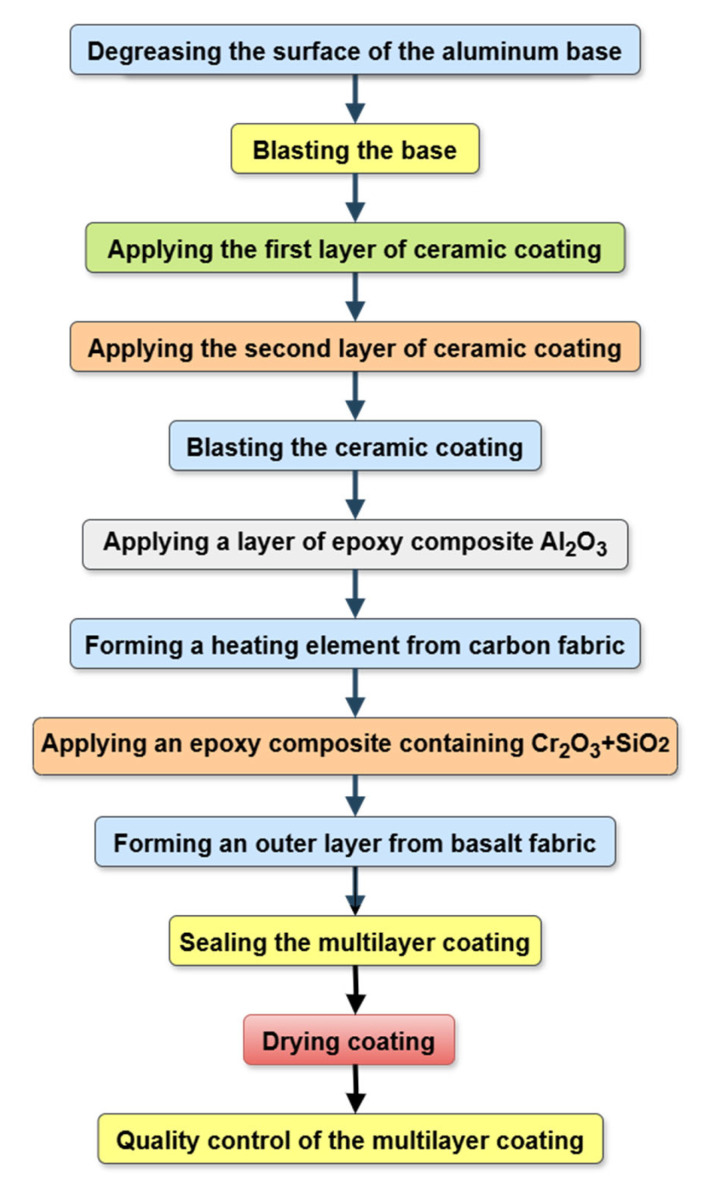
Structural diagram of the technological process for forming multilayer epoxy composite coatings formed on metal–ceramic spraying on an aluminum substrate.

**Figure 4 materials-18-03620-f004:**
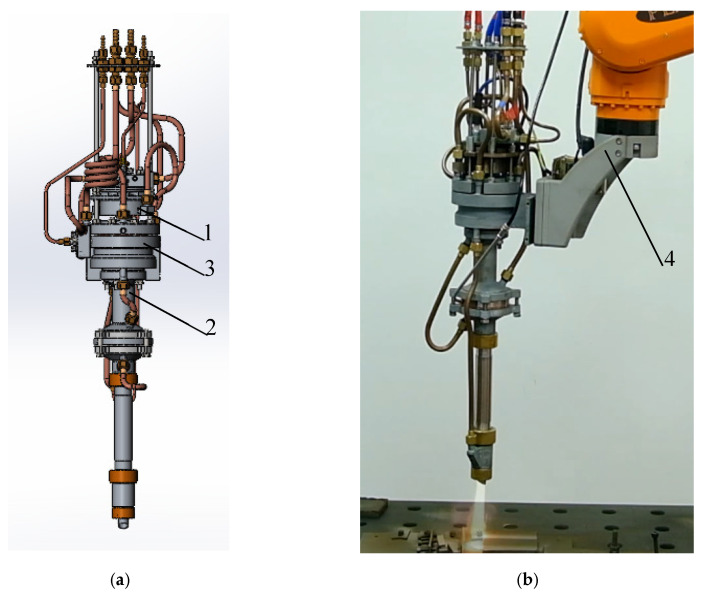
Model of a device for high-speed multi-chamber cumulative detonation spraying (**a**) and general view of this device mounted on a manipulator during detonation-gas spraying of a coating (**b**): 1—nozzle chamber; 2, 3—main and cylindrical chambers; 4—three-axis manipulator.

**Figure 5 materials-18-03620-f005:**
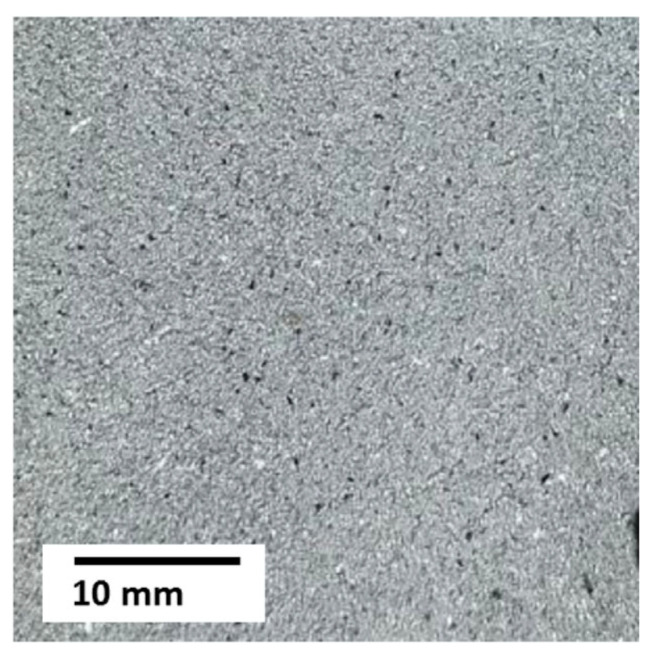
Photograph of an aluminum substrate that was subjected to abrasive blasting with corundum.

**Figure 6 materials-18-03620-f006:**
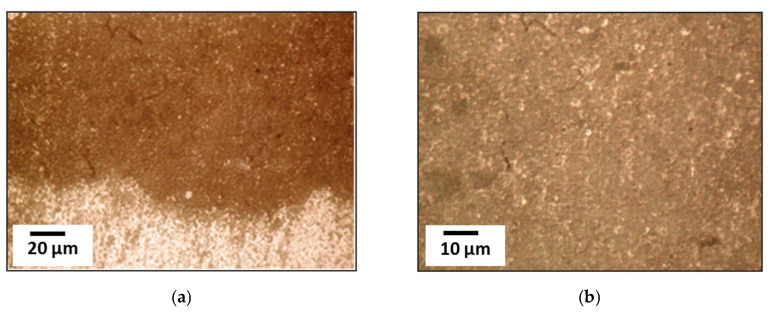
Microstructure of the coating (Al_2_O_3_ + 5% Al) in the fusion zone, ×500 magnification (**a**), and the first layer, ×1000 magnification (**b**).

**Figure 7 materials-18-03620-f007:**
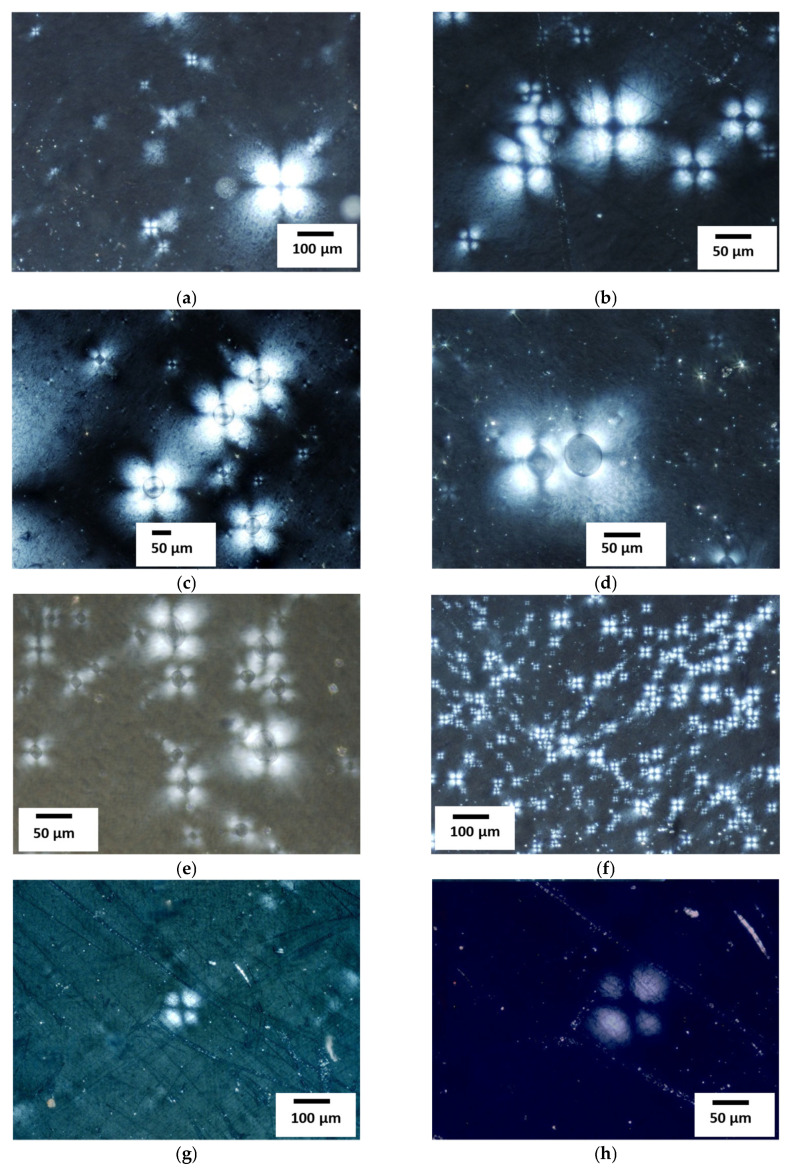
Outer surface layers (OSL): (**a**,**c**,**e**)—when Cr_2_O_3_ is introduced; (**b**,**d**,**f**)—Al_2_O_3_; (**g**,**h**)—SiO_2_; (**a**,**b**,**g**,**h**)—individual particles, (**c**–**f**)—OSL overlap and their convergence.

**Figure 8 materials-18-03620-f008:**
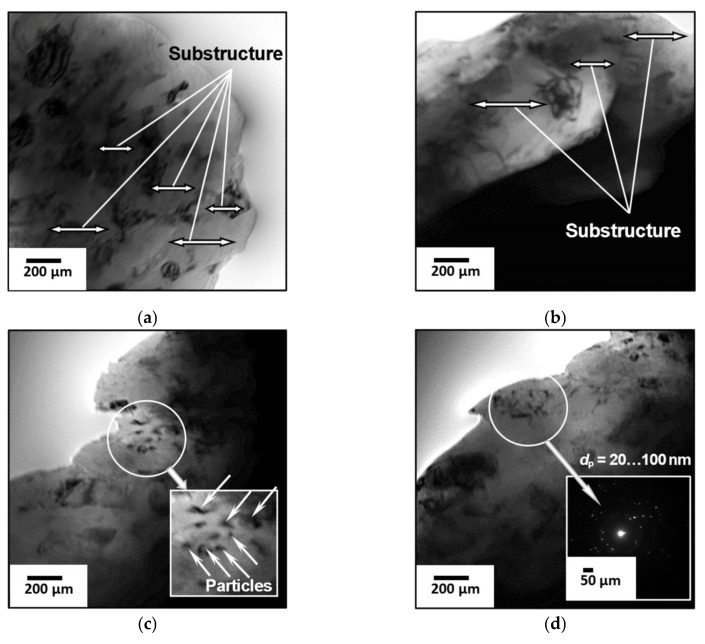
Substructure (**a**,**b**) and nanoparticles of phases (**c**,**d**) in the material of the first coating layer (Al_2_O_3_ + 5% Al), the fusion zone (F/L) of the coating with the substrate, and in the substrate material ((**a**–**d**)—×52,000; (**e**,**f**)—×35,000).

**Figure 9 materials-18-03620-f009:**
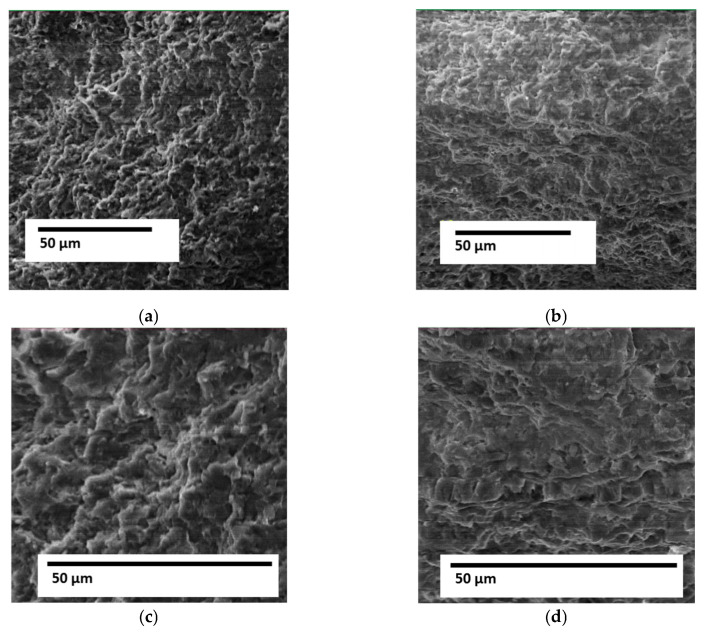
Fracture surface patterns of the Al_2_O_3_ + 5% Al coating layer: (**a**,**b**)—×1200; (**c**,**d**)—×2400.

**Figure 10 materials-18-03620-f010:**
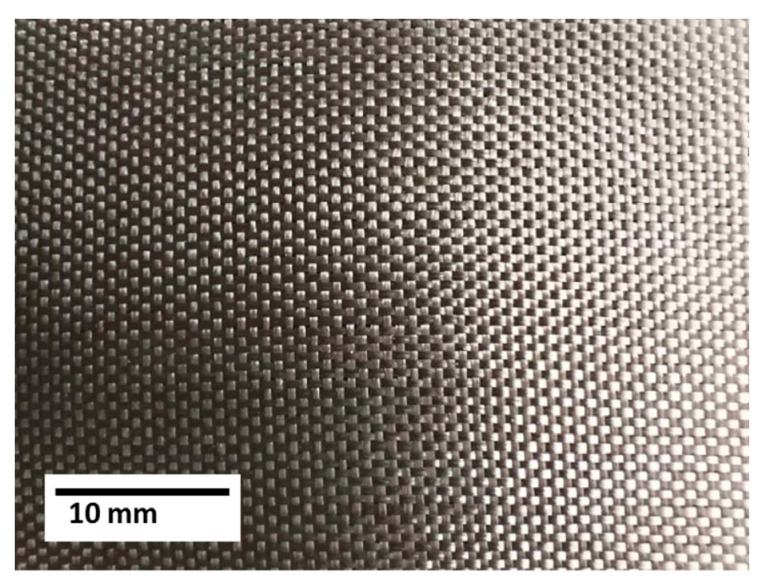
Photograph of the outer layer (basalt fabric) of a multilayer coating.

**Table 1 materials-18-03620-t001:** Parameters of the dislocation structure and indicators of structural strengthening of the material of the first layer (Al_2_O_3_ + 5% Al) of a multilayer coating.

Thin Structure Parameters		Strengthening, MPa	
*d*_s_, nm	100–600	Subgrain (Δσ_S_)	100–875
*d*_p_, nm	10–120	Dispersion (Δσ_D.H._)	884–1256
λ_p_, nm	15–60		
ρ (coating), cm^–2^	(2–3) × 10^9^	Dislocation (Δσ_D_)	124–152
ρ (F/L), cm^–2^	(4–5) × 10^10^		196–215

## Data Availability

The original contributions presented in this study are included in the article. Further inquiries can be directed to the corresponding authors.

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
