# Peer review of "Features of the Structure of Layered Epoxy Composite Coatings Formed on a Metal-Ceramic-Coated Aluminum Base"

_materials, 2025, doi:10.3390/ma18153620_

Round 1
Reviewer 1 Report
Comments and Suggestions for Authors
In this work, a multilayered coating on a parabolic antenna was designed and prepared. The coating includes two aluminum oxide layers by the method of detonation high-speed spraying and epoxy coatings with different oxide fillers. The topic and content are interesting. However, major revisions are necessary before considering publication.
- The abstract needs to be rewritten. It is strange that the Abstract only introduced the work of preparing the first two layers of coatings (Al2O3 + 5% Al).
- The part of the introduction can be more concise.
- The concept of “sputtering” is different from that of “spraying.”
- For the deposition of the first two layers of coatings (Alâ‚‚O₃ + 5% Al), why is the porosity of these two coatings much different (0.5–0% vs. 20–25%)?
- The structure of the prepared coating is quite complicated. It is necessary to explain the function of each layer.
- It will be good to show the pictures of the actual prepared samples. It is necessary to discuss whether the developed coating process is suitable for practical applications.
- In conclusion, it is stated that It is proved that the method of detonation high-speed spraying provides the formation of a sub-grain structure with a size of 100–300 nm and phase particles with a size of 10–120 nm and their uniform distribution in the coating material. It is necessary to characterize the second-phase particles and analyze how they formed.
Author Response
Response to Reviewer 1 Comments
NOTE: Text crossed out and highlighted in yellow should be removed from the article;
Text highlighted in green has been added to the article.
1. Summary |
||
Dear Esteemed Reviewer, thank you very much for taking the time to review this manuscript, thank you so much for your constructive fruitful comments. The comments you provided helped a lot improving our manuscript. Please find the detailed responses below and the corresponding revisions highlighted in the re-submitted files.
|
||
2. Questions for General Evaluation |
Reviewer’s Evaluation |
|
Does the introduction provide sufficient background and include all relevant references? |
Can be improved |
|
Is the research design appropriate? |
Can be improved |
|
Are the methods adequately described? |
Yes |
|
Are the results clearly presented? |
Can be improved |
|
Are the conclusions supported by the results? |
Can be improved |
|
Are all figures and tables clear and well-presented? Can be improved
- Point-by-point response to Comments and Suggestions for Authors
Comments 1: In this work, a multilayered coating on a parabolic antenna was designed and prepared. The coating includes two aluminum oxide layers by the method of detonation high-speed spraying and epoxy coatings with different oxide fillers. The topic and content are interesting. However, major revisions are necessary before considering publication.
Response 1: Thank you very much for recognizing our efforts. Your comments are essential and have improved the quality of our manuscript. We have taken all your comments into account.
Comments 2: The abstract needs to be rewritten. It is strange that the Abstract only introduced the work of preparing the first two layers of coatings (Al2O3 + 5% Al).
Response 2: The authors appreciate your suggestions and make appropriate changes, which can be found in the revised manuscript. The abstract has been rewritten and supplemented.
Abstract: Difficult extreme operating conditions of parabolic antennas under precipitation and sub-zero temperatures require the creation of effective heating systems. The purpose of the research is to develop a multilayer coating containing two metal-ceramic layers and layers of epoxy composites, carbon fabric and an outer layer of basalt fabric, which allows for effective heating of the antenna, and to study the properties of this coating. The multilayer coating was formed on an aluminum base that was subjected to abrasive jet processing. The first and second metal-ceramic layers Al2O3 + 5% Al, which were applied by high-speed multi-chamber cumulative detonation spraying (CDS), respectively, provide maximum adhesion strength to the aluminum base and high adhesion strength to the third layer of the epoxy composite containing Al2O3. On this not-yet-polymerized layer of epoxy composite containing Al2O3, a layer of carbon fabric (impregnated with epoxy resin) was formed, which serves as a resistive heating element. On top of this carbon fabric, a layer of epoxy composite containing Cr2O3 and SiO2 was applied. Next, basalt fabric was applied to this still not-yet-polymerized layer. Then, the resulting layered coating was compacted and dried. To study the multilayer coating, X-ray analysis, light and raster scanning microscopy, and transmission electron microscopy were used. The thickness of the coating layers and microhardness were measured on transverse microsections. The adhesion strength of the metal-ceramic coating layers to the aluminum base was determined by both bending testing and peeling using the adhesive method. It has been established that CDS provides the formation of metal-ceramic layers with a maximum fraction of lamellae and a microhardness of 7900–10520 MPa. In these metal-ceramic layers, a dispersed subgrain structure, a uniform distribution of nanoparticles, and a gradient-free level of dislocation density are observed. Such a structure prevents the formation of local concentrators of internal stresses, thereby increasing the level of dispersion and substructural strengthening of the metal-ceramic layers' material. The formation of materials with a nanostructure increases their strength and crack resistance. The effectiveness of using aluminum, chromium, and silicon oxides as nanofillers in epoxy composite layers has been demonstrated. The presence of structures near the surface of these nanofillers, which differ from the properties of the epoxy matrix in the coating, has been established. Such zones, specifically the outer surface layers (OSL), significantly affect the properties of the epoxy composite. The results of industrial tests showed the high performance of the multilayer coating during antenna heating.
Comments 3: The part of the introduction can be more concise.
Response 3: Thank you for the good recommendation. We agree. The text of 1. Introduction has been shortened, and the references in References have been changed to more relevant ones (See the text of the corrected article).
Comments 4: The concept of “sputtering” is different from that of “spraying.”
Response 4: Thank you for carefully reading the paper and your valuable comment. We agree. The formation of metal-ceramic coatings is carried out by cumulative detonation spraying (CDS).
A correction has been made to the paper: sputtering spraying.
Comments 5: For the deposition of the first two layers of coatings (Alâ‚‚O₃ + 5% Al), why is the porosity of these two coatings much different (0.5–0% vs. 20–25%)?
Response 5: Thank you very much for the constructive comments. The porosity of the first two metal-ceramic coating layers (Alâ‚‚O₃ + 5% Al) differs due to the need to ensure reliable adhesion of the multilayer coating to the aluminum base, the metal-ceramic layers to each other and to the third layer of epoxy composite. First, the first metal-ceramic layer (Alâ‚‚O₃ + 5% Al) with a thickness of 150 μm is applied to the aluminum base by cumulative detonation spraying (CDS), using a fine powder fraction (Alâ‚‚O₃—5–22 μm; Al—60 μm), 10 μm thick and 0–0.5% porosity, which ensures high adhesion strength of this layer to the aluminum base.
Then, a second metal-ceramic layer (Alâ‚‚O₃ + 5% Al) with a thickness of 250 μm is applied by cumulative detonation spraying, using a coarse fraction of powders (Alâ‚‚O₃—60–150 μm; Al—60 μm), creating a guaranteed porosity of 20–25%, which subsequently promotes penetration into the open pores of the epoxy resin and ensures high adhesion strength of the third layer of epoxy composite containing Al2O3 as a filler.
Comments 6: The structure of the prepared coating is quite complicated. It is necessary to explain the function of each layer.
Response 6: Thank you for the valuable constructive recommendation. The aluminum base is subjected to abrasive blasting and immediately proceed to the formation of a multilayer coating. The first metal-ceramic layer Al2O3 + 5% Al, which is applied by cumulative detonation spraying (CDS), has a low porosity (less than 1%) and provides maximum adhesion strength to the aluminum base. The second metal-ceramic layer, which is also sprayed by CDS with high adhesion strength to the first layer of the metal-ceramic layer and guaranteed porosity (20–25%), provides additional adhesive mechanical strength of adhesion to it of the third layer of epoxy composite containing Al2O3, which also has high electrical insulation characteristics. On this, not yet polymerized, layer of epoxy composite containing Al2O3, a layer of carbon fabric (impregnated with epoxy resin) is formed, which serves as a resistive heating element that converts electrical energy into thermal energy. On top of this carbon fabric, a layer of epoxy composite containing Cr2O3 and SiO2 is applied to ensure high electrical insulation strength. Next, on the not yet polymerized layer of the specified epoxy composite containing Cr2O3 and SiO2, a basalt fabric is applied - the outer layer, which performs the function of protection against ultraviolet radiation and has high electrical insulation characteristics.
Comments 7: It will be good to show the pictures of the actual prepared samples. It is necessary to discuss whether the developed coating process is suitable for practical applications.
Response 7: Thank you for the useful recommendation, which allowed us to improve the illustrative presentation of the article, its understanding and practical application of the obtained results. We have added a photo to the revised paper.
Before the first layer of metal-ceramic coating was applied, the aluminum substrate was subjected to abrasive blasting (Figure 5).
Figure 5. Photograph of an aluminum substrate that was subjected to abrasive blasting with corundum.
The developed technological process for forming a multilayer coating is suitable for practical application. To check the workability, the multilayer coating was formed using the developed technology (Figure 3) on the back side of a parabolic antenna made of aluminum alloy and used for its heating. The results of industrial tests of a parabolic antenna with the developed multilayer coating showed its high workability (for 9 years of operation). During a visual inspection of the multilayer coating, using a magnifier with a 7-fold magnification, no defects were found in this multilayer coating, so these tests are continuing.
Comments 8: In conclusion, it is stated that It is proved that the method of detonation high-speed spraying provides the formation of a sub-grain structure with a size of 100–300 nm and phase particles with a size of 10–120 nm and their uniform distribution in the coating material. It is necessary to characterize the second-phase particles and analyze how they formed.
Response 8: Thank you for the valuable constructive recommendation. The formation of nanocluster-sized particles in the metal-ceramic layers of a multilayer coating occurs due to the action of high energies during cumulative detonation spraying on Al2O3 + 5% Al powder particles, which hit a solid, stationary aluminum base at high speed, which contributes to the nucleation and growth of the specified phase particles.
4. Response to Comments on the Quality of English Language |
Point 1: The English is fine and does not require any improvement. |
Response 1: Dear Esteemed reviewer, thank you so much for fruitful comments. The language of the manuscript is thoroughly checked and and the errors are removed carefully. |
5. Additional clarifications |
There is no any other clarifications we would like to provide to the journal Editor and Reviewers. |

Reviewer 2 Report
Comments and Suggestions for Authors……nanostructured with high adhesion, cohesion and resistance to cracks. How do you know that they are nanostructured… what measures did you take to guarantee nanostructured materials?
.. phase particles (nanoparticles) with a size of 10–120 nm….. if they exceed 120 nm they are not nanoparticles
In Figure 1, the measurement scale used is not clear, the same as in Figure 8.
Supplement Suggestion: Further investigation into how the specific composition of the nanofillers (Al2O3, Cr2O3, SiO2) and their exact concentrations influence the degree and topology of cross-linking within these OSL2. This could include advanced studies of
Spectroscopy (FTIR, Raman) to characterize the chemical bonds and the degree of curing in the OSL.â–ª
Atomic force microscopy (AFM), â–ªThermal analysis (DSC, DMA) to investigate the glass transition temperature (Tg).
Given the critical influence of Outer Surface Layers (OSLs) and the impact of surface modification of nanofillers on electrical and mechanical properties, what are the molecular and nanometric mechanisms by which the surface chemistry of nanofillers (Al2O3, Cr2O3, SiO2) directs the topology and density of cross-linking in epoxy OSLs, and how does this controlled microstructure of OSLs correlate with the long-term durability and dielectric stability of multilayer coating under thermomechanical fatigue?
How can the coupled phenomena of heat transfer, adhesive interaction, and thermocyclic fatigue degradation across layer interfaces be modeled and optimized, and what is the optimal design strategy to balance the controlled porosity of the ceramic layers with the thermophysical efficiency and overall mechanical resilience of the coating over its expected service life?
Given that the coating functions as a resistive heating element under extreme conditions and its controlled porosity (5–20%) affects both mechanical adhesion and thermophysical properties, how do the authors quantify and optimize the design of this layer and interfaces to ensure the long-term durability of the system and prevent its localized thermomechanical degradation, ensuring uniform and efficient heat dissipation throughout its useful life?
How is this porosity optimized to prevent localized thermomechanical degradation and ensure uniform and efficient heat dissipation throughout the life of the system?

English is not my first language
Author Response
Response to Reviewer 2 Comments
NOTE: Text crossed out and highlighted in yellow should be removed from the article;
Text highlighted in green has been added to the article.
1. Summary |
||
Dear Esteemed Reviewer, thank you very much for taking the time to review this manuscript, thank you so much for your constructive fruitful comments. The comments you provided helped a lot improving our manuscript. Please find the detailed responses below and the corresponding revisions highlighted in the re-submitted files.
|
||
2. Questions for General Evaluation |
Reviewer’s Evaluation |
|
Does the introduction provide sufficient background and include all relevant references? |
Yes |
|
Is the research design appropriate? |
Can be improved |
|
Are the methods adequately described? |
Can be improved |
|
Are the results clearly presented? |
Can be improved |
|
Are the conclusions supported by the results? |
Yes |
|
Are all figures and tables clear and well-presented? Can be improved
- Point-by-point response to Comments and Suggestions for Authors
Response: Thank you very much for recognizing our efforts. The comments you provided are significant, which improved the quality of our manuscript. We have addressed all of your concerns.
Comments 1: ……nanostructured with high adhesion, cohesion and resistance to cracks. How do you know that they are nanostructured… what measures did you take to guarantee nanostructured materials?
Response 1: Thank you for your valuable comment. As a result of the research conducted by the method of transmission electron microscopy, during the study of the fine structure on the lumen, it was found that a subgrain substructure of the nanoscale type is formed in the coating material. The formation of nanocluster-sized particles in the metal-ceramic layers of the multilayer coating occurs due to the action of high energies during cumulative detonation sputtering (CDS) on Al2O3 + 5% Al powder particles, which hit a solid immobile aluminum base at high speed, which contributes to the nucleation and growth of the specified phase particles.
Comments 2:.. phase particles (nanoparticles) with a size of 10–120 nm….. if they exceed 120 nm they are not nanoparticles.
Response 2: Thank you for your valuable comment. Yes. We agree. Particles larger than 100 nm. Are not nanoparticles.
Comments 3: In Figure 1, the measurement scale used is not clear, the same as in Figure 8.
Response 3: Thank you. We agree with the Reviewer's valid comment. The quality of Figure 1 and Figure 8 has been improved, the measurement scale has been made more visible.
Comments 4: Supplement Suggestion: Further investigation into how the specific composition of the nanofillers (Al2O3, Cr2O3, SiO2) and their exact concentrations influence the degree and topology of cross-linking within these OSL2. This could include advanced studies of Spectroscopy (FTIR, Raman) to characterize the chemical bonds and the degree of curing in the OSL.â–ª Atomic force microscopy (AFM), â–ªThermal analysis (DSC, DMA) to investigate the glass transition temperature (Tg).
Response 4: Thank you for the valuable additional suggestion. In the future, we plan to conduct in-depth studies of the processes and topological parameters during the crosslinking of epoxy composites. The use of the recommended research methods will allow for a deeper understanding of the processes of formation of multilayer coatings. Since epoxy resin emits volatile substances, the study of the above phenomena is associated with the need to develop and improve experimental research methods.
Comments 5: Given the critical influence of Outer Surface Layers (OSLs) and the impact of surface modification of nanofillers on electrical and mechanical properties, what are the molecular and nanometric mechanisms by which the surface chemistry of nanofillers (Al2O3, Cr2O3, SiO2) directs the topology and density of cross-linking in epoxy OSLs, and how does this controlled microstructure of OSLs correlate with the long-term durability and dielectric stability of multilayer coating under thermomechanical fatigue?
Response 5: Thank you very much for the constructive comments. The introduction of nanodispersed fillers Al2O3, as well as Cr2O3 and SiO2) into the composition of the epoxy composite affects the formation of OSL due to the physical interaction between the solid surface of the filler and the binder. On the surface of these fillers there are active centers (exchangeable electrons, OH groups, defects in the crystal structure), which interact with the macromolecules of the epoxy binder. In this case, the macromolecules are presented in the form of domains. Depending on the number of active centers, the layer located near the solid surface of the filler interacts due to the emergence of physical nodes. Such nodes are stable up to the glass transition temperature of the epoxy matrix. Processing of OSL images around dispersed particles in polarized light is characterized by a brightness gradient. Brightness determines the degree of crosslinking in OSL. The binder material in the OSL state has increased mechanical characteristics during cyclic loading. Migration of physical units during such loading will increase the resistance to the formation of micro- and subsequently macrocracks, thus increasing the crack resistance, long-term strength and dielectric stability of the epoxy composite, and accordingly the service life of the multilayer coating. The outer surface layers (OSL) differ in mechanical and dielectric characteristics, and the parameters of such layers do not change during thermomechanical loading. Our previous studies [85] have established an increase in the mechanical strength of materials that include material in the OSL state. The electrical strength of such materials increases from 80 to 120 kV per millimeter of epoxy composite thickness.
Comments 6: How can the coupled phenomena of heat transfer, adhesive interaction, and thermocyclic fatigue degradation across layer interfaces be modeled and optimized, and what is the optimal design strategy to balance the controlled porosity of the ceramic layers with the thermophysical efficiency and overall mechanical resilience of the coating over its expected service life?
Response 6: Thank you very much for the fruitful comments. The strategy for designing a multilayer coating consisted in a rational choice, a combination of physical and mechanical properties of functional layers and their strong connection with each other. Since the research is conceptual, pioneering, and the optimization of geometric and physical parameters of a multilayer coating requires significant efforts and a lot of time, in this study, complex optimization was not carried out, due to the emergence of an urgent need to solve a technical problem - the creation of a parabolic antenna heating system in a short period of time. The authors plan to carry out such multi-criteria optimization in their further research on multilayer coatings.
The aluminum base is subjected to abrasive jet processing and immediately proceed to the formation of a multilayer coating. The first metal-ceramic layer Al2O3 + 5% Al, which is applied by cumulative detonation spraying CDS, has a low porosity (less than 1%) and provides maximum adhesion strength to the aluminum base. The second metal-ceramic layer, which is also applied by CDS, with high adhesion strength to the first layer of the metal-ceramic layer and guaranteed porosity (20–25%), provides additional adhesive mechanical strength of adhesion to it of the third layer of epoxy composite containing Al2O3, which also has high electrical insulation characteristics. On this, not yet polymerized, layer of epoxy composite containing Al2O3, a layer of carbon fabric (impregnated with epoxy resin) is formed, which serves as a resistive heating element that converts electrical energy into thermal energy. On top of this carbon fabric, a layer of epoxy composite containing Cr2O3 and SiO2 is applied to ensure high electrical insulation strength. Next, on the still unpolymerized layer of the specified epoxy composite containing Cr2O3 and SiO2, a basalt fabric is applied - the outer layer, which performs the function of protection against ultraviolet radiation and has high electrical insulation characteristics.
Based on methodological experiments, it was established that:
- the first metal-ceramic layer Al2O3 + 5% Al, which is sprayed by CDS, with low porosity (less than 1%) provides maximum strength of its adhesion to the aluminum base;
- with a porosity of 20–25% of the second metal-ceramic layer (Al2O3 + 5% Al) and the content of Al2O3 in the third layer of the epoxy composite as a filler, high adhesion strength of the specified layers is ensured. The choice of Al2O3 as a filler was due to the fact that both metal-ceramic layers (Al2O3 + 5% Al) also contain aluminum oxide in their composition, i.e., the alignment of the linear expansion coefficients of these layers is ensured. In addition, the third epoxy composite layer has high electrical insulation capacity.
- the presence of aluminum in the composition of the first and second metal-ceramic layers (Al2O3 + 5% Al), which has high thermal conductivity, ensures stable heat transfer from the carbon fabric layer, which serves as a resistive heating element, to the aluminum parabolic antenna. The outer layer of basalt fabric performs the function of protection against ultraviolet radiation and has high electrical insulation characteristics.
During thermal cycling, the adhesion strength of metal-ceramic layers is 0.5–0.6 of the cohesive strength of the coating itself. The adhesion strength of epoxy composite materials is several times lower than the adhesion strength of metal-ceramic layers of the coating. Despite the phenomena of heat transfer, adhesive interaction, thermal cycling, the high operability of the multilayer coating has been experimentally established, which has been operated for more than 9 years on experimental and industrial samples of parabolic antennas without delamination and cracking. In this study, optimization of the multilayer coating was not carried out. This will be the subject of our future research.
Comments 7: Given that the coating functions as a resistive heating element under extreme conditions and its controlled porosity (5–20%) affects both mechanical adhesion and thermophysical properties, how do the authors quantify and optimize the design of this layer and interfaces to ensure the long-term durability of the system and prevent its localized thermomechanical degradation, ensuring uniform and efficient heat dissipation throughout its useful life?
Response 7: Thank you very much for the useful comments. The porosity of less than 1% of the first metal-ceramic layer (Al2O3 + 5% Al) was chosen, from the condition of ensuring its reliable mechanical adhesion to the aluminum base, and the porosity of 20–25% of the second metal-ceramic layer (Al2O3 + 5% Al) was chosen, from the condition of ensuring reliable mechanical adhesion to this metal-ceramic layer of the third layer of epoxy composite containing Al2O3. Heat is effectively removed from the multilayer coating into the body of the working part of the parabolic antenna, which is made of aluminum alloy, which has high thermal conductivity, and dissipates this heat through the mirror (reflective) surface into the environment. Also, part of the heat is removed through the epoxy composite layer containing Cr2O3 and SiO2, and the outer layer, which is made of basalt fabric, into the environment. This design of the multilayer coating provides uniform and effective heat dissipation during heating of the parabolic antenna.
It was experimentally established that for 9 years of operation of the parabolic antenna there is no thermomechanical degradation of the multilayer coating. Heating and uniformity of the thermal field distribution are provided by the heating element made of carbon fabric itself. Heat transfer to the working surface of the parabolic antenna is carried out to remove ice, snow and frost from its working mirror (reflecting) surface of the parabolic antenna during its heating. In this case, the parabolic antenna is heated and enters the operating mode. In this study, optimization of the multilayer coating was not carried out. This will be the subject of our future research.
Comments 8: How is this porosity optimized to prevent localized thermomechanical degradation and ensure uniform and efficient heat dissipation throughout the life of the system?
Response 8: Thank you for the constructive comment. The porosity of less than 1% of the first metal-ceramic layer (Al2O3 + 5% Al) was chosen to ensure its reliable mechanical adhesion to the aluminum substrate, and the porosity of 20–25% of the second metal-ceramic layer (Al2O3 + 5% Al) was chosen to ensure reliable mechanical adhesion to the first metal-ceramic layer and the third layer of epoxy composite containing Al2O3. In this study, optimization of this porosity to prevent localized thermomechanical degradation and ensure uniform and effective heat dissipation throughout the entire service life of the system was not carried out. This will be the subject of our future research.
Response 9: Thank you for your careful review of the article. Item 3. 2% – These are our proceedings of the 2023 IEEE 13th International Conference Nanomaterials: Applications and Properties, NAP 2023, Bratislava, Slovakia, 10 – 15 September 2023 [96].
4. Response to Comments on the Quality of English Language |
Point 1: The English is fine and does not require any improvement. |
Response 1: Dear Esteemed reviewer, thank you so much for fruitful comments. The language of the manuscript is thoroughly checked and and the errors are removed carefully. |
5. Additional clarifications |
There is no any other clarifications we would like to provide to the journal Editor and Reviewers. |

Reviewer 3 Report
Comments and Suggestions for Authors
The paper deals with composite coatings for parabolic antennas. The authors analized the sample by Xray, TEM, posority, and electrical measurement.
The article is interesting but a little unclear and difficult to understand. In particular, some things need to be rearranged.
Please add in the abstract detail the measurements carried out.
Line53. Add which type are the most common?
Figure 2. What is the 5—heating layer composed of?
Line 297.How did you measure the coating thickness?
Line 300.The range of hardness values is very high. Explain why and indicate the standard deviation of the measurements.
Figure 5. Please and in the figures the scale bar.
Figure 7. Please add visible scale bar
Table-1. Also in this case the values dispersion range is very big. Please add explanation
Since we are talking about coatings and adhesion, it would be necessary to do scratch test.
Author Response
Response to Reviewer 3 Comments
NOTE: Text crossed out and highlighted in yellow should be removed from the article;
Text highlighted in green has been added to the article.
1. Summary |
||
Dear Esteemed Reviewer, thank you very much for taking the time to review this manuscript, thank you so much for your constructive fruitful comments. The comments you provided helped a lot improving our manuscript. Please find the detailed responses below and the corresponding revisions highlighted in the re-submitted files.
|
||
2. Questions for General Evaluation |
Reviewer’s Evaluation |
|
Does the introduction provide sufficient background and include all relevant references? |
Can be improved |
|
Is the research design appropriate? |
Not applicable |
|
Are the methods adequately described? |
Can be improved |
|
Are the results clearly presented? |
|
|
Are the conclusions supported by the results? |
Can be improved |
|
Are all figures and tables clear and well-presented? Can be improved
- Point-by-point response to Comments and Suggestions for Authors
Comments 1: The paper deals with composite coatings for parabolic antennas. The authors analized the sample by Xray, TEM, posority, and electrical measurement.
The article is interesting but a little unclear and difficult to understand. In particular, some things need to be rearranged.
Response 1: Thank you very much for recognizing our efforts. The comments you provided are significant, which improved the quality of our manuscript. We have addressed all of your concerns.
Comments 2: Please add in the abstract detail the measurements carried out.
Response 2: The authors appreciate your suggestions and make appropriate changes, which can be found in the revised manuscript. The abstract has been rewritten and supplemented.
Abstract: Difficult extreme operating conditions of parabolic antennas under precipitation and sub-zero temperatures require the creation of effective heating systems. The purpose of the research is to develop a multilayer coating containing two metal-ceramic layers and layers of epoxy composites, carbon fabric and an outer layer of basalt fabric, which allows for effective heating of the antenna, and to study the properties of this coating. The multilayer coating was formed on an aluminum base that was subjected to abrasive jet processing. The first and second metal-ceramic layers Al2O3 + 5% Al, which were applied by high-speed multi-chamber cumulative detonation spraying (CDS), respectively, provide maximum adhesion strength to the aluminum base and high adhesion strength to the third layer of the epoxy composite containing Al2O3. On this not-yet-polymerized layer of epoxy composite containing Al2O3, a layer of carbon fabric (impregnated with epoxy resin) was formed, which serves as a resistive heating element. On top of this carbon fabric, a layer of epoxy composite containing Cr2O3 and SiO2 was applied. Next, basalt fabric was applied to this still not-yet-polymerized layer. Then, the resulting layered coating was compacted and dried. To study the multilayer coating, X-ray analysis, light and raster scanning microscopy, and transmission electron microscopy were used. The thickness of the coating layers and microhardness were measured on transverse microsections. The adhesion strength of the metal-ceramic coating layers to the aluminum base was determined by both bending testing and peeling using the adhesive method. It has been established that CDS provides the formation of metal-ceramic layers with a maximum fraction of lamellae and a microhardness of 7900–10520 MPa. In these metal-ceramic layers, a dispersed subgrain structure, a uniform distribution of nanoparticles, and a gradient-free level of dislocation density are observed. Such a structure prevents the formation of local concentrators of internal stresses, thereby increasing the level of dispersion and substructural strengthening of the metal-ceramic layers' material. The formation of materials with a nanostructure increases their strength and crack resistance. The effectiveness of using aluminum, chromium, and silicon oxides as nanofillers in epoxy composite layers has been demonstrated. The presence of structures near the surface of these nanofillers, which differ from the properties of the epoxy matrix in the coating, has been established. Such zones, specifically the outer surface layers (OSL), significantly affect the properties of the epoxy composite. The results of industrial tests showed the high performance of the multilayer coating during antenna heating.
Comments 3: Line 53. Add which type are the most common?
Response 3: The author appreciates your suggestions and makes appropriate changes, which can be found in the revised manuscript. Added which types of fillers are the most common (See the text of the corrected paper):
…To ensure the creation of highly effective composites, fillers obtained using new technologies are used: [48,49,50].. carbon nanotubes [48], multilayer structures of carbon hybrid polymer nanocomposites [49], graphene oxide modified with 4,4'-oxydianiline obtained by covalent grafting reaction [50], etc. …
Comments 4: Figure 2. What is the 5—heating layer composed of?
Response 4: Thank you very much for the useful comments. The aluminum base is subjected to abrasive blasting and immediately begins to form a multilayer coating on it. The first metal-ceramic layer Al2O3 + 5% Al, which is applied by cumulative detonation spraying (CDS), has a low porosity (less than 1%) and provides maximum adhesion strength to the aluminum base. The second metal-ceramic layer, also applied by CDS, with high adhesion strength to the first layer of the metal-ceramic layer and guaranteed porosity (20–25%), provides additional adhesive mechanical adhesion strength to it of the third layer of epoxy composite containing Al2O3, which also has high electrical insulation characteristics. On this, not yet polymerized, layer of epoxy composite containing Al2O3, a layer of carbon fabric (impregnated with epoxy resin) is formed, which serves as a resistive heating element that converts electrical energy into thermal energy. On top of this carbon fabric, a layer of epoxy composite containing Cr2O3 and SiO2 is applied to ensure high electrical insulation strength. Next, on the not yet polymerized layer of the specified epoxy composite containing Cr2O3 and SiO2, a basalt fabric is applied - the outer layer, which performs the function of protection against ultraviolet radiation and has high electrical insulation characteristics.
Comments 5: Line 297.How did you measure the coating thickness?
Response 5: Thank you for your valuable comment. The thickness of the coating layers applied to the aluminum substrate was measured on transverse microsections of the multilayer coating by the microscopic (optical) method (ISO 1463), with an accuracy of 1 μm, using a measuring optical-digital instrumental microscope IMCL-100x50 (LLC VTP "ASMA-PRYLAD", Svetlovodsk, Ukraine). The total thickness of the multilayer coating was measured with an accuracy of 5 μm using a micrometer MK-25 with a graduation of 0.01 mm and a measurement range of 0–25 mm (Mikrotech, Kharkiv, Ukraine). The thickness of the carbon and basalt fabric was also measured using this micrometer MK-25.
Comments 6: Line 300.The range of hardness values is very high. Explain why and indicate the standard deviation of the measurements.
Response 6: Thank you very much for the useful comments. The ranges of microhardness values ​​of the first and second metal-ceramic layers Al2O3 + 5% Al differ: (HV = 8900–10520 MPa, layer 1); (HV = 7900–10250 MPa, layer 2). This is due to the fact that the composite materials of the specified metal-ceramic layers consist of a soft aluminum matrix Al and a hard filler - aluminum oxide Al2O3, which are characterized by different hardness values, respectively, and have different grain size and porosity. The accuracy of hardness measurement was ≅5%, the standard deviation of the measurement results was about 15–17%.
Comments 7: Figure 5. Please and in the figures the scale bar.
Response 7: Thank you. We agree with the Reviewer's valid comment. The quality of Figure 5 has been improved; the measurement scale has been made more visible.
Comments 8: Figure 7. Please add visible scale bar
Response 8: Thank you. We agree with the Reviewer's valid comment. The quality of Figure 7 has been improved, the measurement scale has been made more visible.
Comments 9: Table-1. Also in this case the values dispersion range is very big. Please add explanation.
Response 9: Thank you for the valuable comment of the Reviewer. The range of scattering of values ​​(Table 1. Parameters of the dislocation structure and indicators of structural strengthening of the material of the first layer (Al2O3 + 5% Al) of the multilayer coating.) is quite large, since it is determined by the sizes of subgrains, the sizes of dispersed particles of phases, the distances between particles and the density of dislocations. We determined these sizes using transmission electron microscopy. Next, we determined the strengthening (Subgrain, Dispersion, Dislocation), using the obtained data according to the method described in the paper [9,13,88].
Comments 10: Since we are talking about coatings and adhesion, it would be necessary to do scratch test.
Response 10: Thank you for the valuable constructive remark of the Reviewer. The authors of the article are aware of the Scratch Test Standard [ISO 2819:2017 Metallic coatings on metallic substrates - Electrodeposited and chemically deposited coatings - Review of methods available for testing adhesion]: "2.8 Scribe and grid test. Using a hardened steel scribe which has been ground to a sharp 30° point, two parallel lines are scribed at a distance apart of about 2 mm. In scribing the two lines, enough pressure shall be applied to cut through the coating to the base metal in a single stroke. If any part of the coating between the lines breaks away from the base metal, the coating shall be deemed to have failed the test. One variation of the test is to draw a square with a grid of 1 mm side and observe whether the coating peels from the base metal within this area." We did not use this test because the ceramic layer of the multilayer coating is very hard and it was difficult to scratch through to the aluminum substrate. In addition, this test only gives a qualitative assessment of the strength of adhesion to the substrate.
Therefore, in our study:
The adhesion strength of the ceramic coating layers to the aluminum base was determined by the adhesive method. A layer of metal-ceramic coating was formed on the end surface of cylindrical samples with a diameter of 0.01 m, made of aluminum alloy, after abrasive blasting by cumulative detonation spraying (CDS). Two such samples with metal-ceramic coatings were coaxially glued with VK-31 glue, which was polymerized at a temperature of 175 °C for 90 minutes under a load of 1 MPa. Peel tests were performed using self-centering grips on a Universal testing machine AGS-X, 10N–10kN (Shimadzu Corporation, Kyoto, Japan). The adhesion value of the coating was calculated by the well-known formula σ = 4F/(πd2). Here, F is the peel force, N; π is a constant number (≅3.14), d is the diameter of the cylindrical sample, 0.01 m. Based on the results of five tests of the samples, the arithmetic mean value of the adhesion strength of the coating to the base was determined, the value of which was approximately 80 MPa. In this case, destruction by the adhesive occurred.
4. Response to Comments on the Quality of English Language |
Point 1: The English is fine and does not require any improvement. |
Response 1: Dear Esteemed reviewer, thank you so much for fruitful comments. The language of the manuscript is thoroughly checked and and the errors are removed carefully. |
5. Additional clarifications |
There is no any other clarifications we would like to provide to the journal Editor and Reviewers. |

Round 2
Reviewer 1 Report
Comments and Suggestions for Authors
The revised mansucript can be accepted now.
Author Response
Thank you for the quality review.
Reviewer 3 Report
Comments and Suggestions for Authors
The authors answered the questions completely. Only a few things need to be corrected.
Line 24. Why layer and layer?
Please add the scale bar on figure 5.
